# The Enhancing Effects of 10% PBS Washout of Holocene Minerals on HuIFN-αN3 Inducing Capacity of NDV ZG1999HDS or Sendai virus (Cantell Strain)

**DOI:** 10.3390/life12030414

**Published:** 2022-03-12

**Authors:** Bratko Filipič, Lidija Gradišnik, Adriana Pereyra, Gordan Mršić, Marjan Andrašec, Hrvoje Mazija

**Affiliations:** 1CIETO (Croatian Institute for Experimental and Translational Oncology), Koledinečka 03, 10040 Zagreb, Croatia; marijan.andrasec1@zg.ht.hr (M.A.); mazija@vef.unizg.hr (H.M.); 2Institutes of Biomedical Sciences, Medical Faculty, University of Maribor, Taborska 8, 2000 Maribor, Slovenia; lidija.gradisnik@um.si; 3AMEU–ECM Maribor, Slovenska 17, 2000 Maribor, Slovenia; 4MEDEX D.o.o., Linhartova cesta 49a, 1000 Ljubljana, Slovenia; adriana.pereyra@medex.si; 5Ministry of the Interior, General Police Directorate, Forensic Centre “Ivan Vučetić” Ilica 305, 10000 Zagreb, Croatia; gordan.mrsic@gmail.com

**Keywords:** virus NDV ZG1999HDS, Sendai virus (Cantell strain), interferon induction, 10% PBS washout, Holocene minerals, enhancement

## Abstract

Different strains of Newcastle disease viruses (NDV) or Sendai viruses (SV) are used to induce the production of human leukocyte multi subtype interferon-alpha (HuIFN-αN3). Their inducing capacity can be enhanced in different ways. One includes 10% PBS washout of Holocene minerals (HM). The presented study aims to compare the HuIFN-αN3 inducing capacity of NDV ZG1999HDS or SV (Cantell strain) strain in vitro, and to evaluate the enhancing effect of 10% PBS washouts of HM on both viruses. The NDV strains’ ZG1999HDS interferon inducing capacity (483.23 ± 4.5 pg/mL) was similar to that of the SV (Cantell strain) (584.16 ± 5.9 pg/mL). It was shown that the HuIFN-αN3 inducing capacity of the strain of NDV ZG1999HDS can be strongly enhanced with 10% PBS washout of HM to 3818.21 ± 41.9 pg/mL and 4790.34 ± 33.5 pg/mL with SV (Cantell strain), u. The RP-HPLC analyses of such HuIFN-αN3 induced with the strain of NDV ZG1999HDS show the difference to SV (Cantell strain) induced HuIFN-αN3 in the absence of subtype α14 and the lower level of the subtype α1. The possible ways of such enhancement were also studied and it was postulated that the Fe^2+^ ions from 10% PBS washouts of HM, while stimulating the reactive oxygen species (ROS) and nitric oxide (NO) formation, activate the transcription factor NF- κB and consequently the production of HuIFN-αN3.

## 1. Introduction

Interferons (IFNs) are multifunctional glycoproteins/proteins that are produced and released by host cells as a response to the presence of different pathogens, such as viruses, bacteria, parasites, or tumor cells [1]. They are also able to trigger the protective defenses of the immune system that decisively contribute to the elimination of pathogens or tumors [2]. There are three major classes of IFNs, designated as Type I, II, and III. Type I includes: IFNs-α, β, ω, ε and κ, Type II includes IFN-γ [3,4], while Type III includes IFN λ [5]. Human IFN-α (HuIFN-αN3) consists of a family of approximately 22 structurally related proteins, which are the products of 14 different genes. They are located on chromosome 9. Three of them are glycosylated, with an approximate molecular weight of 17,500–27,000. Each of them is composed of 165–166 amino acids. The HuIFN-αN3 induction in the peripheral blood mononuclear cells (PBMC) from human Buffy coats was performed with the strain of NDV ZG1999HDS [6] or SV (Cantell strain) [7]. For the HuIFN-αN3 induction in human PBMC cells, the hem agglutinin-neuraminidase (HN) membrane glycoprotein of the NDV virus is needed [8,9]. Among various NDV strains that can induce the HuIFN-αN3, the new strain of NDV, ZG1999HDS was recently isolated, patented and characterized [10,11]. In 2013, it was deposited in the [12] Collection National de Cultures de Microorganisms (CNCM) and in the year 2014 in the Gene Bank [13]. The virus was isolated from the lung tissue of broiler chickens sufferings a respiratory disease, as it was described by Biđin and Mazija [14]. The virus was fully sequenced and genetically characterized [15], while its phylogenetic and genome sequence analyses showed that the strain of NDV ZG1999HDS belongs to the genotype II of class II of NDVs, that are closely related to the NDV strains La Sota and Hitcher B1.

The Holocene era denotes a period from 9.560 to 9.300 B.C. starting with the withdrawal of the Pleistocene glaciations. During that time, the Holocene sands of the Drava River near the city of Koprivnica (Croatia), containing the fairly uniform, mostly silicate-bearing HM, were formed. When grained, they show unusual biological/microbiological activity, such as antifungal activity against *Peronospora* sp. and *Phytoftora* sp. that was previously found by Filipič et al. [16]. The same authors found that the 10% PBS washout of grained HM enhances the HuIFN-αN3 antiproliferative and pro-apoptotic activity against Colon cancer carcinoma (CaCo-2) cells in vitro [17].

The present experiments were performed to determine the HuIFN-αN3 inductive capacity of the strain of NDV ZG1999HSD in comparison to SV (Cantell strain) and the possibility of their enhancement with the 10% phosphate buffer saline (PBS) washouts of HM.

## 2. Material and Methods

### 2.1. Viruses

The strain of NDV ZG1999HDS was obtained from Prof. emeritus Hrvoje Mazija, CEO of the CIETO (Croatian Institute for Experimental and Translational Oncology) Koledinečka 03, 10040 Zagreb, Croatia. The SV (Cantell strain) was kindly provided by Dr Eugen Šooš, PhD. Both viruses were multiplied in Specific Pathogen Free (SPF) chicken embryos and concentrated by lyophilization. The EID_50_ determined in SPF chicken embryos was 2.0 × 10^7^ for the strain of NDV ZG1999HDS and 1.0 × 10^7^ for SV (Cantell strain).

### 2.2. 10% PBS Washout of HM and Analysis of 10% PBS Washout of HM Crystals

10% PBS washout was prepared from grained HM. The PBS was added, and the 10% PBS washout was shaken extensively and then centrifuged at 1700 RPM for 20 min. The supernatant was filtered through the 0.2 μm syringe filters and filled into 10.0 mL sterile plastic tubes that were stored at −20.0 °C. Such supernatant was used as a “10% PBS washout of HM” in all experiments. The grained HM were chemically analyzed in the Croatian Geological Survey, Department of Mineral Resources in Zagreb (Croatia) using the method described by Smith et al. [18]. The analyses’ results are shown in Table 1.

For analytic purposes, the 10% PBS washout of HM solution was crystallized, and these crystals were analyzed by scanning electron microscopy (SEM) by the method ESM/EDX (Automated analysis for GSR-Gun Shot Residua, Nanoscience Instruments, Phoenix, AZ, USA). The Scanning microscopes Tescan-Mira3Feg (Tescan, Brno, Czech Republic) with EDX detector using Program QuantaxEDS (Bruckner, Karlsruhe, Germany) were used. Samples for analysis were taken on aluminum stubs (carrier of the sample), (Micro to Nano, Ethaarlem, Netherlands) with adhesive trace strip, and GSR software (Global Software Resources, Pleasanton, CA, USA) was applied for analysis. The results of these analyses are shown in Table 2.

### 2.3. The Isolation of Peripheral Blood Mononuclear Cells (PBMC) from Human Buffy Coats

Human Buffy coats of different blood donors obtained from the Blood Transfusion Centre in Ljubljana (Slovenia) were combined and centrifuged at 1.700 RPM for 20 min at 4 °C. To the erythrocytes, lymphocytes, macrophages and granulocytes, nine parts of 0.83% ammonium chloride were added. The erythrocyte lyses was performed at 4 °C and took 15 to 20 min. Then, the cell suspension was centrifuged at 2.500 RPM for 20 min at 4 °C and the supernatant was removed, while white cell sediments were resuspended in the PBS containing 1% of glucose. The percentage of living or dead cells was determined by Trypan blue staining. The isolated PBMC cells were resuspended in the HuIFN-αN3 induction medium prepared according to the method of Jerker’s and Olovson [19] in a concentration of 2.0 × 10^7^ cells/mL + 5% of Human Serum Albumin (HSA).

### 2.4. The Determination of the Haemmaglutination (HA) Activity of the Strain of NDV ZG1999HDS and SV (Cantell Strain)

The HA activity of the virus was determined by the use of 1% solution of chicken erythrocytes as described by Atabekov [20].

### 2.5. The Induction of HuIFN-αN3 with the Strain of NDV ZG1999HDS or SV (Cantell Strain)

To the 5 cm Petri dishes containing 3 mL of PBMCs (2 × 10^7^ cells/mL) cells suspension in the HuIFN-αN3 induction medium, 3.2, 6.4, 9.6 and 100 HA/mL of the strain of the NDV ZG1999HDS were added. In the case of SV (Cantell strain), the same amount of virus was added. The Petri dishes were incubated for 18 h at 37 °C at 5% CO_2_. All experiments were performed in triplicate in the three to four separated experiments. In the separate experiments, before the viral induction of the strain of NDV ZG1999HDS or SV (Cantell strain), the priming with 100 IU/mL of HuIFN-αN3 was performed for two hours at 36 °C.

### 2.6. HuIFN-αN3 Induction Enhancement Experiments

To the 100 HA/mL of the NDV ZG1999HDS or SV, 10% PBS washout of HM was added, and induction was performed. In the combined experiment 100 IU/mL of HuIFN-αN3 priming and 10% PBS washout of HM was added to the 100 HA/mL of the NDV ZG1999HDS or SV (Cantell strain), respectively. All experiments were performed in triplicate in the three to four separated experiments.

### 2.7. HuIFN-αN3 Monoclonal ELISA, (“Platinum ELISA”)

The amount of induced HuIFN-αN3 (pg/mL) was determined by the Human IFN ELISA kit [21] (Platinum ELISA from eBioscience, Vienna, Austria). In the assay, the international HuIFN-αN3 standard was used (Human IFN-α “Platinum ELISA“ BMS216/BMS 216 TEN, Affymetrix, eBioscience, 3420 Central Expressway, Santa Clara, CA, USA). The assay was performed in accordance with the manufacturer’s instructions, with the final reading on the ELISA reader at 620 nm and calculating the pgs (picograms) of HuIFN-αN3/mL.

### 2.8. HuIFN-αN3 RP-HPLC Analysis

The HuIFN-αN3 subtype composition was analyzed by reverse-phase high-performance liquid chromatography (RP-HPLC). The HPLC column was provided by Phenomenex, Aeris PEPTIDE, column 3.6 μm XB-C18, 250 mm × 4.6 mm. Different HuIFN-α samples (natural and recombinant) with approximately one million antiviral (AV) units/mL in a volume of 20 to 40 μL were applied to the column and eluted with a linear gradient of Solvent A = water + 0.1% of TFA and Solvent C = Acetonitrile + 0.1% TFA for 20 min with a flow rate of 0.8 mL/min. and pressure of 139 to 140 bar. The course of RP-HPLC chromatography of different IFN samples is shown in the Table 3. The temperature of the column was 40 °C. The absorbance was monitored at 214 and 280 nm. HuIFN-αN3 species in different IFNs compositions were separated according to their relative hydrophobicity using RP-HPLC as was described by Alm et al. [22] and Wang et al. [23].

### 2.9. NO Assay

The concentration of stable NO, which is the end product of nitric oxide, is present in the supernatant of treated or untreated human PMBCs cells (2.0 × 10^7^ cells/mL). It was measured by the method based on the Griess reaction described by More and Pai [24]. The absorbance at 550 nm was measured in a microtiter plate reader and the standard curve for NO was prepared by the use of 10–100 μM sodium nitrites in distilled water.

### 2.10. Lysozyme Determination

The amount of lysozyme was determined in accordance with the method developed by Nash et al. [25]. One thousand Colony Forming Units (CFU) of *Streptococcus pyogenes* in 200 μL of 10 mM of Potassium phosphate buffer (pH 7.4) and 200 μL of sample or 1.0, 10.0 and 100.0 μg/mL of Lysozyme were separately added to the 1.6 mL of Mueller–Hinton (MH)–Broth (pH 6.5) and incubated overnight at 37 °C. On the next day, the OD was measured at 595 nm. The amount of Lysozyme (μM/mL) was calculated in comparison to the bacterial OD after 24 h.

### 2.11. Statistics

All of the treatments were performed in triplicate and repeated three to four times. The average values of standard deviation (SD) were recorded. The obtained data were analyzed with the *t*-test. The Stat graphics Stratus online statistics software (www.statgraphicsstratus.com, accessed on 13 January 2022) from Stat point Technologies Inc., Warrenton, VA, USA was used. Statistically significant were differences with the *p* values: * *p* ≤ 0.05, ** *p* ≤ 0.01, *** *p* ≤ 0.001, **** *p* ≤ 0.0001.

### 2.12. The Plan of the Experiments

During the experiments shown in Figure 1, HuIFN-αN3 was induced by 100 HA/mL of NDV ZG1999HDS or SV (Cantell strain) with or without 10% PBS washout of HM and with or without priming with 100 IU/mL of HuIFN-αN3. The HuIFN-αN3 induction was performed as follows: To the PBMCs cells + adsorbed macrophages, the 100 IU/mL of HuIFN-αN3 primed for 120 min at 37 °C was added. This was followed by the addition of 100 HA/mL of ZG1999HDS or SV (Cantell strain) for 60 min at 35 °C. Then, the 10% PBS washout of HM was added to enhance the HuIFN-αN3 induction. The samples were incubated at 35 °C for 24 h. The incubation, cells (PBMCs + adsorbed macrophages) was then centrifuged at 1.700 RPM for 15 min. The decanted supernatants, which were filtered through the 0.2 μm syringe filters, were analyzed as follows: (1) RP-HPLC profile of treated and untreated cells. (2) The ELISA assay of the HuIFN-αN3 content. (3) Assay of NO. (4) Assay of the Lysozyme content.

## 3. Results

### 3.1. HuIFN-αN3′s Induction Capacity of NDV ZG1999HDS or SV (Cantell Strain)

The results of the HuIFN-αN3 induction experiments in the human PBMCs with the strain of NDV ZG1999HDS or SV (Cantell strain) are shown in the Figure 2. In the case of NDV strain, ZG1999HDS induction of the HuIFN-αN3, a relatively high amount (429.11 ± 5.3 pg/mL) was obtained with the use of only 3.2 HA units/mL of the strain of NDV ZG1999HDS. The comparison between 3.2 HA units/mL and 6.4 HA units/mL show the relatively strong decrease of HuIFN-αN3 induction to the 283.49 ± 7.6 pg/mL at 6.4 HA units/mL. In the case of SV (Cantell strain) at 3.2 HA units/mL the 275.22 ± 6.8 pg/mL of HuIFN-αN3 was obtained. At 6.4 HA units/mL the decrease to the 258.11 ± 6.2 pg/mL of HuIFN-αN3 was found. The decrease is much higher in the case of NDV ZG1999HDS than on SV (Cantell strain). The mechanism of such decrease is recently unknown. The highest amounts of HuIFN–αN3 (pg/mL) were obtained with 100 HA/mL of the strain of NDV ZG1999HDS 483.23 ± 4.5 pg/mL or 100 HA/mL of SV (Cantell strain) 584.16 ± 5.9 pg/mL.

### 3.2. HuIFN-αN3 Enhancement Induction Experiments with 10% PBS Washout of HM

#### 3.2.1. HuIFN-αN3 Enhancement Induction Experiments with 10% PBS Washout of HM without Priming with 100 IU/mL of HuIFN-αN3

The results of HuIFN-αN3 enhancement induction experiments with 10% PBS washout of HM without priming with 100 IU/mL of HuIFN-αN3 are presented in the Table 4. The results without priming with 100 IU/mL of HuIFN-αN3 show the relatively high enhancement in the amount of HuIFN-αN3. The most interesting are the data of the enhancement with 10% PBS washout of HM of 3.6 IU/mL of NDV ZG1999HDS (3045.32 ± 30.2 pg/mL of huIFN-αN3), which are higher than that obtained after enhancement of 3.6 IU/mL of SV (Cantell strain). This is exactly the opposite situation of the enhancement of the 100 IU/mL of NDV ZG1999HDS or SV (Cantell strain), where the higher value (4790.34 ± 33.5 pg/mL of HuIFN-αN3) was found after enhancement of induction with the SV (Cantell strain). The possible mechanisms of such differential enhancement are unknown, even the role of low–doses (3.6 HA/mL) of NDV ZG1999HDS should not be neglected.

Similarly, in the enhancement experiments with the 10% PBS washout of HM the level of the 4790.34 ± 33.5 pg/mL of the HuIFN-αN3 was obtained. It is interesting to note, that the priming with the 100 units of HuIFN-αN3/10^7^ of PMBC did not increase the level of produced HuIFN-αN3 after the addition of 10% PBS washout of HM or SV (Cantell strain). The same effect was obtained using 100 IU/mL priming with HuIFN-αN3/10^7^ of PMBCs and induction with the strain of NDV ZG1999HDS + 10% PBS washout of HM.

#### 3.2.2. HuIFN-αN3 Enhancement Induction Experiments with 10% with PBS Washout of HM and Priming with 100 IU/mL of HuIFN-αN3

The results of the HuIFN-αN3 induction enhancement are shown in Table 5. The most fascinating is the data of the priming experiments by the 100 IU/mL of HuIFN-αN3 for two hours at 37 °C and then adding the 100 HA/mL of the NDV strain ZG1999HDS or SV (Cantell strain) and separately 10% PBS washout of HM.

The priming destroys the induction enhancement of the 10% PBS Washouts of HM alone with the 100 HA/mL either the strain of NDV ZG1999HDS or SV (Cantell strain), which deserves further analysis. Similarly, in the enhancement experiments with the 10% PBS washout of HM the level of the 4790.34 ± 33.5 pg/mL of the HuIFN-αN3 was obtained. It is interesting to note that the priming with the 100 units of HuIFN-αN3/10^7^ of PMBCs did not increase the level of produced HuIFN-αN3 after the addition of 10% PBS washout of HM or SV (Cantell strain). The same effect was obtained using 100 IU/mL priming with HuIFN-αN3/10^7^ of PMBCs and induction with the strain of NDV ZG1999HDS + 10% PBS washout of HM.

### 3.3. Macrophage’s Activation: NO Assay and Lysozyme Determination

The results regarding the amount of NO are shown in Table 6. In both cases 100 HA/mL of NDV ZG1999HDS (Figure 3) or SV (Cantell strain) (Figure 4) 50 mM of FeCl_2_ caused a decrease in NO level. The data regarding the amount of Lysozyme are shown in the same table. The differences can be seen between NDV ZG1999HDS, in which 50 mM FeCl_2_ increases the amount of Lysozyme while in the case of SV (Cantell strain) 50 mM of FeCl_2_ decreases the amount of Lysozyme.

### 3.4. The RP-HPLC Analyses of the Strain of NDV ZG1999HDS versus SV (Cantell Strain) Induced Interferon (HuIFN-αN3)

The separation of different HuIFN-αN3 subtypes in the samples was achieved by the Acetonitrile concentration. The least hydrophobic interferon subtypes were eluted as early peaks, and the most hydrophobic interferon subtypes were eluted as later. As the standards, the following human recombinants interferon’s were used: HuIFN-αA, HuIFN-α2a and HuIF-α2b. The chromatograms of these at 280 nm and the chromatograms at 280 nm of the Russian HuIFN-αN3 (NDV induced) are shown. The position of different HuIFN-αN3 subtypes was determined in accordance with the 214 nm chromatogram at WO 99/64440 [26] and US 6,309,862 [27] and are shown in Figure 5 and Figure 6. The protein profile (280 nm) was compared to the HuIFN-αN3 profile of different subtypes at 214 nm.

Components of the strain of the NDV ZG1999HDS induced HuIFN-αN3 shown in Figure 5 are the subtypes: α1, α2, αA, and α2b. The most important are the subtypes: α1 and α2 as the main biologically active components of the virus-induced HuIF-αN3, or more precisely its relative ratio in the HuIFN-αN3 preparation. To quantify different HuIFN-αN3 subtypes: α2, and α1 increase (values of mAU relative units), αA remain unchanged, while α2b decrease (value of mAU relative units) (Figure 5B,C).

The main components of the SV HuIFN-αN3, shown in Figure 6, are natural IFN subtypes: α1, α2, αA, αb and α14. Similarly, as in the case of the strain of NDV ZG1999HDS induced HuIFN-αN3 preparation, the most important are the subtypes α1 and α2 as the most active components of the preparation of HuIFN-αN3. The relative ratio between α1 and α2 (values of mAU relative units) is most important for IFN’s biological activity.

HuIFN-αN3 interferon subtypes in different samples (natural and recombinant) are separated according to their relative hydrophobicity by the RP-HPLC [28]. The separation of different HuIFN-αN3 subtypes in the samples was achieved by increasing acetonitrile concentration. The least hydrophobic interferon subtypes were eluted as early peaks and the most hydrophobic interferon subtypes, which were eluted as later.

## 4. Discussion

The results of the HuIFN-αN3 single induction experiments in the human PBMCs with the strain of NDV ZG1999HDS or SV (Cantell strain) are shown in Figure 2. In the case of the strain of NDV ZG1999HDS induction of the HuIFN-αN3, high amounts of 429.11 ± 5.3 pg/mL were obtained with the use of only 3.2 HA units/mL of the strain of NDV ZG1999HDS. Thus, obtained IFNs amounts were comparable with the HuIFN-α3 inducing capacity of the SV (Cantell strain), in which the highest amount in a single experiment was obtained with the 100 HA/mL when 584.16 ± 5.9 pg/mL of HuIFN-αN3 were found. NDV ZG1999HDS HuIFN-αN3 inducing capacity is comparable to the HuIFN-αN3 inducing capacity of the SV (Cantell strain) [28]. The similar data were obtained with the different strains of NDVs. So, the NDV strain Ulster on human PMBCs from Human Buffy coats, induced about 450.34 ± 3.2 pg/mL of HuIFN-αN3 that was obtained with 25 HA units/mL of the virus, as described by Židovec and Mažuran [29].

The RP-HPLC Analyses of the Strain of NDV ZG1999HDS versus SV (Cantell strain) induced Interferon (HuIFN-αN3) show that the HuIFN-αN3 interferon subtypes in different samples (natural and recombinant) are separated according to their relative hydrophobicity by the RP-HPLC [30,31]. The components of the strain of the NDV ZG1999HDS induced HuIFN-αN3 shown in Figure 5 are the subtypes: α1, α2, αA, and α2b. The most important are the subtypes: α1 and α2 as the main biologically active components of virus-induced HuIF-αN3, or more precisely its relative ratio in the HuIFN-αN3 preparation [32,33,34]. To quantify different HuIFN-αN3 subtypes: α2, and α1 increase (values of mAU relative units), αA remain unchanged, while α2b decrease (value of mAU relative units) (Figure 5B,C). Different types of HuIFN-αN3 inductors differ in induction capacity of the amount sof the IFN natural subtype’s α1, α2, αA and α2b. The highest amount of HuIFN-αN3 (3818.21 ± 41.9 pg/mL) itself was obtained with 10% PBS washout of HM addition to 100 HA/mL of the NDV ZG1999HDS.

The NDV ZG1999HDS belongs to avian *Paramyxoviruses*. As a lentogenic, it belongs to the genotype II of class II of NDV. In the same group are the strains: LaSota, Ulster, and Queensland. All of them show low-to-medium cytotoxicity for chicken embryo fibroblasts (CEF). Most of them are inherently oncolytic and tumor selective, sparing the normal cells. The cytolytic characteristics of the strain of NDV ZG1999HDS were investigated in vitro on tumor cell cultures and in vivo on mice. They were compared with the impact of the strain La Sota. The tumor selectivity of lentogenic NDVs is considered to be due to a defective IFN’s response in tumor cells. Moreover, the NDV ZG1999HDS is a relatively strong inducer of human type I IFNs, more precisely of the HuIFN-αN3, in the PBMCs from human Buffy coats. The 100 HA/mL of the NDV ZG1999HDS mixed with 10% PBS washout of HM can induce the 3818.21 ± 41.9 pg/mL of the HuIFN-αN3. The RP-HPLC profile of the HuIFN-αN3 show subtypes α1, α2, αA and α2b. The predominant components of the Sendai virus-induced HuIFN-αN3, shown in Figure 6, are the natural IFN subtypes α1, α2, αA, αb and α14. Similarly, as in the case of the strain of NDV ZG1999HDS induced HuIFN-αN3 preparation, the most important are the subtypes α1, α2 and α14, as the most active components of the preparation of HuIFN-αN3. The most important is the relative ratio between α1 and α2 (values of mAU relative units). The quantification of different subtypes shows a α2 and α1 increase (values of mAU relative units), while αA, α2b and α14 remain unchanged (Figure 6B,C). Various types of HuIFN-αN3 inductors differ in induction capacity of HuIFN-αN3 subtype’s α1, α2, αA, α2b and α14. The highest amount of HuIFN-αN3 (4790.34 ± 33.5 pg/mL) itself was obtained with 10%PBS washouts of HM addition to 100 HA/mL of SV (Cantell strain).

The SV (Cantell strain) virus is a negative-stranded RNA virus with the ability to induce very large quantities of Type I IFNs, more precisely HuIFN-αN3 in the PBMCs from human Buffy coats. The SV (Cantell strain) induced HuIFN-αN3 is composed of the 14 different natural subtypes and exhibits different antiviral, antiproliferative and imimuno modulatory activity in vitro. With the 100 HA units/mL of SV (Cantell strain) in the PBMCs from human Buffy coats 584.16 ± 5.9 pg/mL of the HuIFN-αN3 can be obtained, which is ¼ more than by 100 HA of the strain of NDV ZG1999HDS. Thus, the HuIFN-αN3 inductive capacity of the NDV ZG1999HDS is comparable to the SV (Cantell strain).

The possible mechanisms of the 10% PBS washouts of HM enhancement of HuIFN-αN3 Induction with NDV ZG1999HDS or SV (Cantell Strain) show that the HuIFN-αN3 inducing capacity of the strain of NDV ZG1999HDS can be enhanced with the 10% PBS washout of HM to the level of 3818.21 ± 41.9 pg/mL. A higher amount of HuIFN-αN3 (4790.34 ± 33.5 pg/mL) was obtained with 10% PBS washout of HM addition to 100 HA/mL of SV (Cantell strain). The mechanism of such enhancement is not clear. The analysis of the 10% PBS washout of HM crystals (Table 2) shows the high amount of Fe^2+^ ions. In this respect, it is possible that the HuIFN-αN3 induction enhancement role was investigated by the addition of 50 mM of FeCl_2_, FeCl_3_ and KCl salts as a control (Table 5). The amount of HuIFN-αN3 (pg/mL) the produced NO (μM/mL) and the Lysozyme (μM/mL) were measured. In the case of FeCl_2_, the lowest amount of HuIFN-αN3 (pg/mL) 25.05 ± 0.4, in the case of NDV ZG1999HDS and 23.0 ± 0.8 in case of SV (Cantell strain) were obtained. When the level of NO (μM/mL) was analyzed, there was a decrease in the case of NDV ZG1999HDS to 6.0 ± 0.47 and in the case of SV (Cantell strain) to 6.9 ± 0.35. This differs from the Lysozyme (μM/mL) determination, specifically in the case of strain of NDV ZG1999HDS, with the increase to 15.1 ± 1.4 and in the case of SV (Cantell strain) in which a decrease of 9.2 ± 0.72 was found.

A possible explanation could be that HuIFN-αN3 induces the deregulation of intracellular Fe^2+^ ions and promotes the deregulations of Iron homeostasis in a macrophage during the systemic infection with the intracellular pathogen *Candida galbrata*, leading to its survival. By engaging JAK1, IFNs-I disturbs the balance of the transcriptional activator NRF2 and repressor BACH1 to induce down regulation of the key iron exporter Fpn1 in macrophages. This leads to enhanced iron accumulation in the phagolysosome and failure to restrict fungal access to iron pools. As a result, *Candida galbrata* acquires iron via the Sit1/Ftr1 iron transporters system, which is facilitating the fungal intracellular replication and immune evasion. Thus, HuIFN-αN3 is a central regulator of iron homeostasis, which can affect infection, and restrict the iron bioavailability, which may offer therapeutic strategies to combat invasive microbe infections [35].

The reports suggest that ROS act as a mediator in signal transduction pathways, in which cells react to the surplus of intracellular ROS with the induction of gene expression of proteins involved in the regulation of the cellular redox state. The transcription factor NF-kB plays an important role in a stress responses. Based on the finding that antioxidant proteins inhibit NF-kB activation, it was suggested that NF-kB activity is regulated by intracellular ROS levels. An important component that participates in the ROS formation via the Fenton reaction is free intracellular iron. Since intracellular iron homeostasis is regulated by ferritin, it was suggested that ferritin might serve as a cytoprotective protein, minimizing oxygen free radical formation by sequestering intracellular iron. This idea is supported by the finding that exposure of cells to inducers of ROS such as hydrogen peroxide and tumor necrosis factor α results in the induction of ferritin synthesis. The observation that the *Mengovirus* leader protein interferes with both the cellular iron homeostasis and the activation of NF-kB provides an explanation for the mechanism by which the leader protein down regulates the antiviral host cell response. Induction of ferritin expression in *Mengovirus*-infected cells will limit the availability of iron for the production of free hydroxyl radicals. As a consequence, NF-kB activation and thereby alpha/beta Interferon expression are suppressed in *Mengovirus*-infected cells. An essential step in this antiviral response is the activation of the double-stranded-RNA-dependent protein kinase (PKR) and subsequent activation of NF-kB-mediated expression of genes such as that for Beta Interferon. The data described show that the *Mengovirus* leader protein interrupts the antiviral host cell response by suppression of NF-kB activation, possibly via interaction with the cellular iron metabolism. Another possibility is that Fe^2+^ ions from 10% PBS washout of HM stimulate the ROS and NO formation, and through this, the activation of NF-κB with the induction of HuIFN-αN3. This would explain the NO decrease, specifically in the case of *Mengovirus* leader proteins, which suppressed the Fe^2+^ activation of NF-kB through the ROS (NO) [36].

In the case of the strain of NDV ZG1999HDS, responsible for interferon induction is the complex hem agglutinin-neuraminidase, and in the case of SV (Cantell strain) the defective-interfering genome [37], they do not inhibit the Fe^2+^ mediated activation of NF-κB through the ROS and NO. It is also possible that Fe^2+^ ions bind to the plasma γ-Globulin fraction and induces interferon [38]. Even the possible role of the Au or different Au-salts, due to their presence in the 10% PBS washout of HM (Table 1) should not be neglected due to their strong immunogenic role.

## 5. Conclusions

In summation, it can be concluded that strain of NDV ZG1999HDS showed the HuIFN-αN3 inducing capacity, similar to that of the 10% PBS washout of HM to the SV (Cantell strain). Its HuIFN-αN3 inducing capacity can be enhanced with the 10% PBS washout of HM to the 3818.21 ± 41.9 pg/mL in comparison to the 100 HA/mL of NDV ZG1999HDS alone, in which 483.23 ± 4.5 pg/mL was obtained, despite its induced HuIFN-αN3 is lacking the natural subtype α14, and have a lower amount of the IFN’s natural subtype α1.

## Figures and Tables

**Figure 1 life-12-00414-f001:**
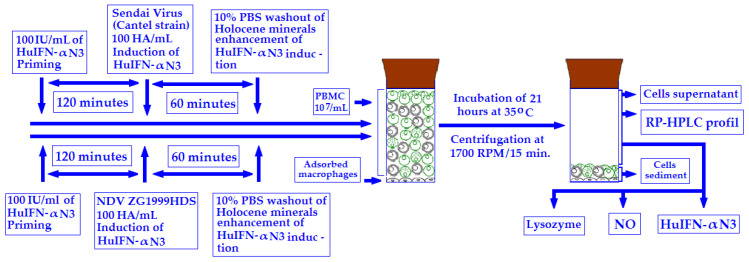
The plan of the experiments.

**Figure 2 life-12-00414-f002:**
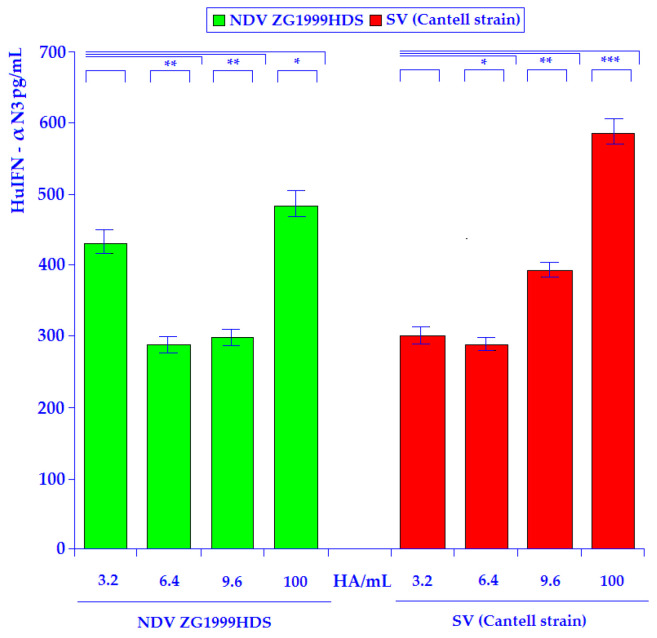
HuIFN-αN3 induction in PBMC by the NDV strain ZG1999HDS or SV (Cantell strain). The experiments were performed in triplicate in three to four separate measurements. Data are presented as means ± standard deviation (SD). Statistically significant were differences with the *p* values: * *p* ≤ 0.05, ** *p* ≤ 0.01, *** *p* ≤ 0.001. One-way ANOVA followed by posttest for NDV ZG1999HDS: The *f*-ratio value is 103.56535. The *p*-value is < 0.00001. The result is significant at *p* < 0.05. One-way ANOVA followed by posttest for SV (Cantell strain): The *f*-ratio value is 165.02672. The *p*-value is < 0.00001. The result is significant at *p* < 0.05.

**Figure 3 life-12-00414-f003:**
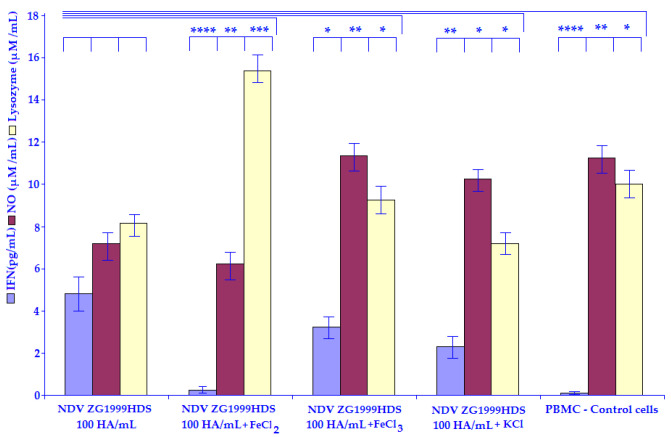
Induction of HuIFN–αN3 (pg/mL), NO (μM/mL) and Lysozyme (μM/mL) by the: NDV ZG1999HDS, NDV ZG1999HDS + 50 mM FeCl_2_, NDV ZG1999HDS + 50 mM FeCl_3,_ NDV ZG1999HDS + 50 mM KCl and PBMCs–control cells. Statistically significant were differences with the *p* values: * *p* ≤ 0.05, ** *p* ≤ 0.01, *** *p* ≤ 0.001, **** *p* ≤ 0.0001.

**Figure 4 life-12-00414-f004:**
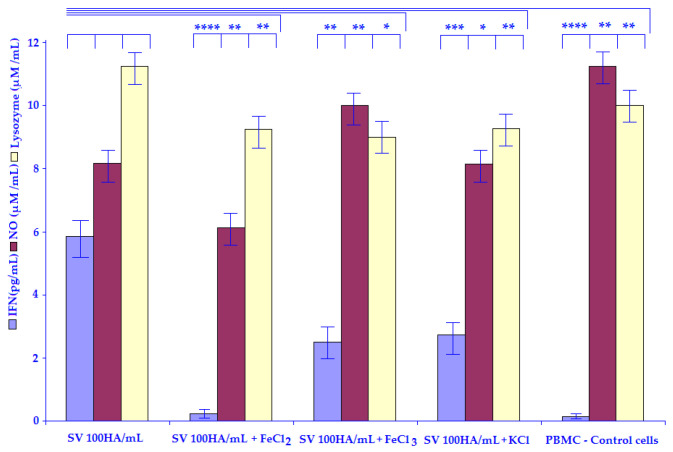
Induction of HuIFN–αN3 (pg/mL), NO (μM/mL) and Lysozyme (μM/mL) by the: SV (Sendai virus) (Cantell strain) 100 HA/mL, SV 100 HA/mL + 50 mM FeCl_2_, SV 100 HA/mL + 50 mM FeCl_3_, SV 100 HA/mL + 50 mM KCl and PBMCs–Control cells. Statistically significant were differences with the *p* values: * *p* ≤ 0.05, ** *p* ≤ 0.01, *** *p* ≤ 0.001, **** *p* ≤ 0.0001.

**Figure 5 life-12-00414-f005:**
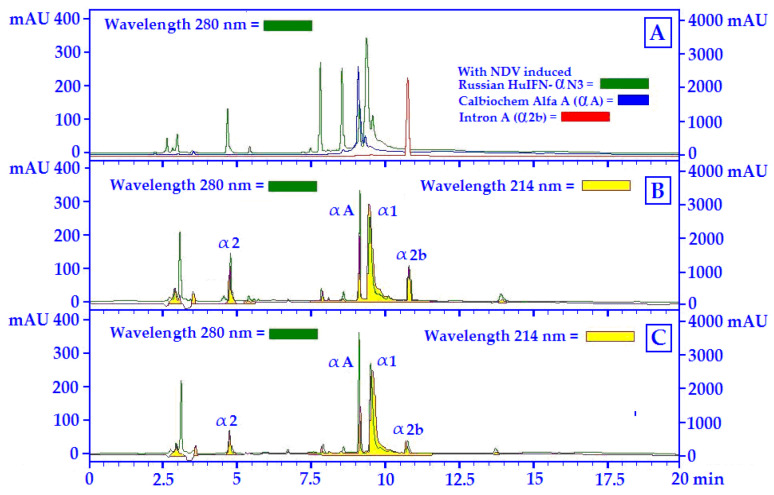
RP-HPLC profiles of the NDV ZG1999HDS induced IFNs: (**A**) Protein profiles of different IFNs at 280 nm; (**B**) Protein profile at 280 nm (

) and IFN profile at 214 nm (
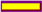
) of HuIFN-αN3 induced with 100 HA/mL of NDV ZG1999HDS; (**C**) Protein profile at 280 nm (

) and IFN profile at 214 nm (
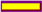
) of HuIFN-αN3 induced with 100 HA/mL NDV ZG1999HDS + 10% PBS washout of HM.

**Figure 6 life-12-00414-f006:**
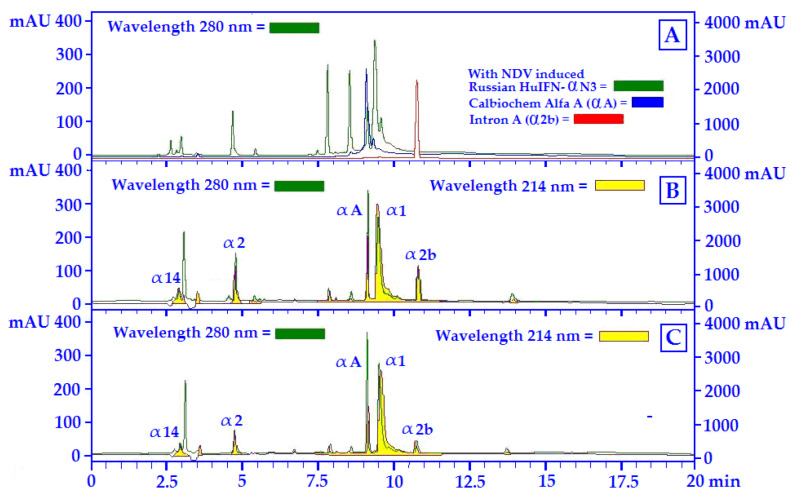
RP-HPLC profiles of the Sendai virus induced IFNs: (**A**) SV, Sendai virus (Cantell strain). Protein profiles of the various IFNs at 280 nm; (**B**) Protein profile at 280 nm (

) and IFN profile at 214 nm (
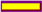
) of HuIFN-αN3 induced with 100 HA/mL of SV; (**C**) Protein profile at 280 nm (

) and IFN profile at 214 nm (
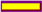
) of HuIFN-αN3 induced with 100 HA/mL SV + 10% PBS washout of HM.

**Table 1 life-12-00414-t001:** The chemical composition of the grained HM from the sands near the Drava River in Koprivnica (Croatia).

Analyte:	UNIT:	MDL:	Sample:	Analyte:	UNIT:	MDL:	Sample:
SiO_2_	%	0.01	88.71	La	PPM	0.1	12.7
Al_2_O_3_	%	0.01	5.37	Pr	PPM	0.02	2.95
Fe_2_O_3_	%	0.04	1.08	Nd	PPM	0.3	11.0
MgO	%	0.01	0.44	Sm	PPM	0.05	2.22
CaO	%	0.01	0.66	Eu	PPM	0.02	0.47
Na_2_O	%	0.01	1.41	Gd	PPM	0.05	2.19
K_2_O	%	0.01	0.92	Tb	PPM	0.01	0.33
TiO_2_	%	0.01	0.18	Dy	PPM	0.05	1.89
P_2_O_5_	%	0.01	0.12	Ho	PPM	0.02	0.42
MnO	%	0.01	0.02	Er	PPM	0.03	1.08
Cr_2_O_3_	%	0.002	0.004	Tm	PPM	0.01	0.18
Ni	%	20.0	<20.0	Yb	PPM	0.05	1.27
Sc	%	1.0	3.0	Lu	PPM	0.01	0.19
Ba	PPM	1.0	154.0	Mo	PPM	0.1	0.4
Be	PPM	1.0	<1.0	Cu	PPM	0.1	5.0
Co	PPM	0.2	2.8	Pb	PPM	0.1	5.8
Cs	PPM	0.1	1.1	Zn	PPM	1.0	17.0
Ga	PPM	0.5	5.4	Ni	PPM	0.1	11.4
Hf	PPM	0.1	3.6	As	PPM	0.5	1.9
Nb	PPM	0.1	5.0	Cd	PPM	0.1	<0.1
Rb	PPM	0.1	36.1	Sb	PPM	0.1	0.1
Sn	PPM	1.0	<1.0	Bi	PPM	0.1	0.3
Sr	PPM	0.5	75.4	Ag	PPM	0.1	<0.1
Ta	PPM	0.1	0.6	Au	PPM	0.5	12.1
Th	PPM	0.2	4.6	Hg	PPM	0.01	0.01
U	PPM	0.1	1.2	Ti	PPM	0.1	<0.1
V	PPM	8.0	11.0	Se	PPM	0.5	<0.5
W	PPM	0.5	1.3	LOI	%	5.1	1.1
Zr	PPM	0.1	131.0	TOT/C	%	0.02	0.12
Y	PPM	0.1	12.2	TOT/S	%	0.2	<0.02
Ce	PPM	0.1	25.2	Summa	%	0.01	99.7

PPM, parts per million; MDL, minimal decimal level; LOI, loss of ignition in sediment; TOT/C, total C analysis; TOT/S, total S analysis.

**Table 2 life-12-00414-t002:** The mineral analysis of the 10% PBS washout of HM crystals.

	10% HM–PBS Crystals	
Analytes	Absolute Values	Relative Values
LaCe	3.0	0.336
Fe^2+^	129.0	14.478
Sb	0	0
Ba	0	0
Pb	1.0	0.112
BaAl	21.0	0.112
BaCaSi	3.0	0.337
SiKCa++	67	7.512
K	68.7	77.104
Au	1.0	0.168
Summa	891	100
Summa absolute	909	

**Table 3 life-12-00414-t003:** The course of RP-HPLC chromatography of different IFN samples.

Step	Time (Minutes)	Solvent A	Solvent C
0	0	91	9
1	3	80	20
2	6	50	50
3	12	50	50
4	15	91	9
5	20	91	9

**Table 4 life-12-00414-t004:** HuIFN-αN3 enhancement induction experiments with 10% PBS washout of HM without the priming with 100 IU/mL of HuIFN-αN3.

Induction with ^(1)^ NDV ZG1999HDS	The Amount of HuIFN-αN3 (pg/mL)	The Amount of HuIFN-αN3 (pg/mL)	Induction with SV ^(2)^ (Cantell Strain)
3.6 HA/mL ^(3)^	435.12 ± 3.9	275.22 ± 6.8	3.6 HA/mL ^(3)^
+10%PBS washout of HM	3045.32 ± 30.2	1925.34 ± 22.6	+10% PBS washout of HM
6.4 HA/mL ^(3)^	283.49 ± 7.6	258.11 ± 6.2	6.4 HA/mL ^(3)^
+10%PBS washout of HM	1698.22 ± 11.3	1806.43 ± 12.9	+10% PBS washout of HM
9.6 HA/mL ^(3)^	295.32 ± 8.6	325.43 ± 3.4	9.6 HA/mL ^(3)^
+10%PBS washout of HM	2065.46 ± 58.2	2275.43 ± 65.3	+10% PBS washout of HM
100 HA/mL ^(3)^	483.23 ± 4.5	584.16 ± 5.9	100 HA/mL ^(3)^
+10%PBS washout of HM	3818.21 ± 41.9	4790.34 ± 33.5	+10% PBS washout of HM

^(1)^ NDV, Newcastle disease virus; ^(2)^ SV, Sendai virus; ^(3)^ HA/mL, hem agglutination units/mL.

**Table 5 life-12-00414-t005:** HuIFN-αN3 enhancement induction experiments with 10% PBS washout of HM with the priming with 100 IU/mL of HuIFN-αN3.

Induction with ^(1)^ NDV ZG1999HDS	The Amount of HuIFN-αN3 (pg/mL)	The Amount of HuIFN-αN3 (pg/mL)	Induction with SV ^(2)^ (Cantell Strain)
NDV ZG1999HDS 100 HA/mL ^(3)^	483.23 ± 4.5	584.16 ± 5.9	SV(Cantell strain) 100 HA/mL
NDV ZG1999HDS 100 HA/mL + 10% PBS washout of HM^(4)^	3818.21 ± 41.9	4790.34 ± 33.5	SV(Cantell strain) 100 HA/mL + 10% PBS washout of HM
HuIFN-αN3 100 IU/10^7^ PBMCs ^(5)^ + NDV ZG1999HDS 100 HA/mL	2695.10 ± 22.4	3447.29 ± 47.3	HuIFN-αN3 100 IU/10^7^ PBMCs + SV(Cantell strain) 100 HA/mL
HuIFN-αN3 100 IU/10^7^ PBMCs + NDVZG1999HDS 100 HA/mL + 10% PBS washout of HM	772.12 ± 9.2	442.24 ± 1.3	HuIFN-αN3 100 IU/10^7^ PBMCs + SV(Cantell strain) 100 HA/mL + 10% PBS washout of HM

^(1)^ NDV, Newcastle disease virus; ^(2)^ SV, Sendai virus; ^(3)^ HA/mL, hem agglutination units/mL; ^(4)^ PBS–HM, 10% PBS washout of the HM; ^(5)^ (HuIFN-αN3 100IU/10^7^ PBMCs), priming with 100 IU/10^7^ PBMCs of HuIFN-αN3.

**Table 6 life-12-00414-t006:** HuIFN-αN3, NO level and Lysozyme amount obtained after the addition of 50 mM FeCl_2_, 50 mM FeCl_3_ or 50 mM KCl to the NDV ZG1999HDS or to SV (Cantell strain).

Samples:	The Amount of HuIFN-αN3 (pg/mL)	The Amount of Nitrite (NO) (μM/mL)	The Amount of Lysozyme (μM/mL)
NDV^(1)^ ZG1999HDS 100HA ^(3)^	483.23 ± 4.5	7.2 ± 1.8	15.0 ± 1.4
NDV ZG1999HDS 100HA ^(3)^ + 50 mM FeCl_2_	25.05 ± 0.4	6.4 ± 0.9	15.4 ± 2.8
NDV ZG1999HDS 100HA ^(3)^ + 50 mM FeCl_3_	320.27 ± 4.9	11.6 ± 3.5	9.4 ± 2.5
NDV ZG1999HDS 100HA ^(3)^ + 50 mM KCl	232.12 ± 8.1	10.16 ± 2.7	7.6 ± 1.8
SV^(2)^ (Cantell strain) 100HA ^(3)^	584.16 ± 5.9	8.6 ± 1.7	11.7 ± 2.4
SV (Cantell strain) 100HA ^(3)^ + 50 mM FeCl_2_	23.0 ± 0.8	6.9 ± 0.35	9.2 ± 0.72
SV (Cantell strain) 100HA ^(3)^ + 50 mM FeCl_3_	250.46 ± 1.6	10.4 ± 2.1	9.6 ± 0.11
SV (Cantell strain) 100HA ^(3)^ + 50 mM KCl	272.33 ± 2.4	8.6 ± 1.4	9.7 ± 1.7
Untreated PBMCs ^(4)^	18.7 ± 3.3	11.6 ± 2.4	10.2 ± 1.4

^(1)^ NDV, Newcastle disease virus; ^(2)^ SV, Sendai virus; ^(3)^ HA/mL, hemagglutination units/mL; ^(4)^ PBMCs, peripheral blood mononuclear cells.

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
