# Peer review of "The Enhancing Effects of 10% PBS Washout of Holocene Minerals on HuIFN-αN3 Inducing Capacity of NDV ZG1999HDS or Sendai virus (Cantell Strain)"

_life, 2022, doi:10.3390/life12030414_

Round 1

Reviewer 1 Report

This paper described the effect of 10% PBS washout of holocene minerals on HuIFN - αN3 inducing capacity of NDV ZG1999HDS or sendai virus (Cantell Strain). Which is meaningful, but whether Newcastle disease virus can duplicate efficiently in PMBC cells is doubtable.

The location of different HuIFNα N3 isoforms determined by reversed-phase high performance liquid chromatography should be verified by intellectual property cooperative experiments and structural electron microscopy.

Viral doses are more convincing in terms of TCI than HA

Author Response

Comments and Suggestions for Authors by Reviewer 1

1.This paper described the effect of 10% PBS washout of holocene minerals on

    HuIFN - αN3 inducing capacity of NDV ZG1999HDS or sendai virus (Cantell

    Strain). Which is meaningful, but whether Newcastle disease virus can duplicate

    efficiently in PMBC cells is doubtable.

ANSWER:

The enhancing effect of 10% PBS washout of Holocene minerals on HuIFN-αN3 inducing capacity of NDV ZG1999HDS or Sendai virus (Cantell strain) was performed as clear cut experiments measured by standard ELISA assay.

2. The location of different HuIFNα N3 isoforms determined by reversed-phase  

     high performance liquid chromatography should be verified by intellectual

     property cooperative experiments and structural electron microscopy.

ANSWER:

HuIFNα N3 subtype composition was determined by reversed-phase high performance liquid chromatography through theirs’ hydrophobic performance. It has not any connection to the isoforms.

Viral doses are more convincing in terms of TCI than HA.

ANSWER:

The HA assay is even old an efficient a valuable assay of Haemagglutinine titer useful in virological experiments.

The relationship between the viral dose (VD) and outcome of infection is expressed by TCID50 . The determination of the viral dose with the 50% endpoint titer using a simple formula. To clarify see the working protocol.

Reviewer 2 Report

In my opinion  the Manuscript ID: life-1536293 is well written and analyzed and is acceptable.

Author Response

Comments and Suggestions for Authors by the Reviewer 2

  1. In my opinion the Manuscript ID: life-1536293 is well written and analyzed and  is acceptable.

ANSWER:

Despite the reviewer’s positive opinion, the finally corrected Manuscript ID: life-1536293 is presented.

Reviewer 3 Report

In the manuscript "the enhancing effects of 10% PBS washout of holocene mine - rals on HuIFN - alpha N3 inducing capacity of NDV ZG1999HDS or sendai virus (Cantrell strain), the authors aim to show the difference of the virus-induced huIFN-aN3 composition. The overall quality of the here presented manuscript is very low. While the experimental design is adequately described, the main idea of the study remains enigmatic. If the authors want to decipher the difference in virus-induced IFN response, and the different antiviral properties of these IFN-alpha subtypes, they should further include in vitro infections experiments in the presence of the different virus strains. Furthermore, the study is suffering from many shortcomings e.g. low readability of the report, or wrong statistic analysis (students t-test is used instead of a One-way ANOVA followed by a post-test, in figure 3 for example).

Author Response

Comments and Suggestions for Authors by the Reviewer 3

Although in general, very interesting and well designed the reviewed manuscript contains plenty of minor errors. My comments is aimed to enhance the clarity of it.

Line 30 – Please select more carefully keywords. Please avoid the duplication of words already used in the article’s title.

Line 62 – please correct “hem agglutinin – neuraminidase” into “hemagglutinin – neuraminidase”

Line 60 - does writing in bold Peronospora sp. and Phytoftora sp. have any special meaning?

Line 64 – why in vitro is written bold ?

Line 65 – present experiment instead of “presented”

Line 105 – Why the acronym PBS is expanded for the second time ? (the first one was in line 78). By the way it was used for the first time in line 62 and this should the place of expansion which should looks like “10% phosphate buffer saline (PBS)”

Line 86 – Holocene minerals are sometimes abbreviated to HM and sometimes not. Please unify it.

Line 108 – please correct “HSA (Human Serum Albumin)” into “human serum albumin (HSA)” .

Line 125 – please correct “mine – rals” into “minerals”

Line 134 - please provide country of origin of Human IFN-α

Line 150 – there is no need to abbreviate nitrite. It is a simple word. Moreover, NO is commonly used for nitric oxide. Therefore it could be misleading.

Line 350 – “Paramyxoviruses” should be written in italic.

Line 317 – I see no reason to use references to figures in Discussion. These should be used in Results only.

Line 318 – why “The Possible Mechanisms” is written in capital letters ?

Line 391, 403, 442, 443  – please correct “Fe++” into “Fe+2

Line 413 – please correct “ROS (Reactive Oxygen Species)” into “Reactive oxygen species (ROS)”

Line 408 - Candida glabrata should be written in italic

Line 436 – Why ROS is expanded for the second time ?

ANSWER:

The corrected version according to the reviewer request is added.

Reviewer 4 Report

In the present paper I carefully reviewed, the Authors aimed to compare the HuIFN-αN3 inducing capacity of the strain of NDV ZG1999HDS or SV (Cantell strain) in vitro, and also to evaluate the enhancing effect of a washout of Holocene minerals with 10% PBS (HM-PBS) on both viruses.

I would like to congratulate Authors for the good-quality of their article, the literature reported used to write the paper, and for the clear and appropriate structure.

The manuscript is well written, presented and discussed, and understandable to a specialist readership.

In general, the organization and the structure of the article are satisfactory and in agreement with the journal instructions for authors. The subject is also adequate with the overall scope of Life.

The work shows a conscientious study in which a very exhaustive discussion of the literature available has been carried out.

The Introduction section provides sufficient background (however, additional references of recently published papers may add value to this section), and the other sections include results clearly presented and analyzed exhaustively.

However, as specific comments, with the aim to further improve the quality of the paper, the Conclusion section could be improved; also, the Authors have to check if all references have been cited in the text.

Moreover, check the style of the references list according to the journal's guidelines. Finally, an overall check of the English language is suggested.

Author Response

The answers to the suggestions of the Reviewer 4

In the present paper I carefully reviewed, the Authors aimed to compare the HuIFN-αN3 inducing capacity of the strain of NDV ZG1999HDS or SV (Cantell strain) in vitro, and also to evaluate the enhancing effect of a washout of Holocene minerals with 10% PBS (HM-PBS) on both viruses.

I would like to congratulate Authors for the good-quality of their article, the literature reported used to write the paper, and for the clear and appropriate structure.

The manuscript is well written, presented and discussed, and understandable to a specialist readership.

In general, the organization and the structure of the article are satisfactory and in agreement with the journal instructions for authors. The subject is also adequate with the overall scope of Life.

The work shows a conscientious study in which a very exhaustive discussion of the literature available has been carried out.

The Introduction section provides sufficient background (however, additional references of recently published papers may add value to this section), and the other sections include results clearly presented and analyzed exhaustively.

However, as specific comments, with the aim to further improve the quality of the paper, the Conclusion section could be improved; also, the Authors have to check if all references have been cited in the text.

Moreover, check the style of the references list according to the journal's guidelines. Finally, an overall check of the English language is suggested.

ANSWER: At first, I wish to thanks for the positive opinions about our article.

As second: All references were cited in the text

As third: An owerall check of English language was performed by AJE (=see the result).

Language Quality Evaluation

Evaluated on 2022-02-06

Upload another document for evaluation

6.6

Language quality evaluation

This paper scored a 6.6/10 which is in the 81st percentile of papers submitted to AJE.

After editing by AJE, papers improve on average to a score of 8.7 and move into the 95th percentile of research papers submitted to journals.

Congratulations! Your document is well written and does not need language editing.

Reviewer 5 Report

Although in general, very interesting and well designed the reviewed manuscript contains plenty of minor errors. My comments aimed to enhance the clarity of it.

Line 30 – Please select more carefully keywords. Please avoid the duplication of words already used in the article’s title.

Line 62 – please correct “hem agglutinin – neuraminidase” into “hemagglutinin – neuraminidase”

Line 60 - does writing in bold Peronospora sp. and Phytoftora sp. have any special meaning?

Line 64 – why in vitro is written bold ?

Line 65 – present experiment instead of “presented”

Line 105 – Why the acronym PBS is expanded for the second time ? (the first one was in line 78). By the way it was used for the first time in line 62 and this should the place of expansion which should looks like “10% phosphate buffer saline (PBS)”

Line 86 – Holocene minerals are sometimes abbreviated to HM and sometimes not. Please unify it.

Line 108 – please correct “HSA (Human Serum Albumin)” into “human serum albumin (HSA)” .

Line 125 – please correct “mine – rals” into “minerals”

Line 134 - please provide country of origin of Human IFN-α

Line 150 – there is no need to abbreviate nitrite. It is a simple word. Moreover, NO is commonly used for nitric oxide. Therefore it could be misleading.

Line 350 – “Paramyxoviruses” should be written in italic.

Line 317 – I see no reason to use references to figures in Discussion. These should be used in Results only.

Line 318 – why “The Possible Mechanisms” is written in capital letters ?

Line 391, 403, 442, 443  – please correct “Fe++” into “Fe+2

Line 413 – please correct “ROS (Reactive Oxygen Species)” into “Reactive oxygen species (ROS)”

Line 408 - Candida glabrata should be written in italic

Line 436 – Why ROS is expanded for the second time ?

Author Response

The answers to the suggestions of Reviewer 5

Although in general, very interesting and well designed the reviewed manuscript contains plenty of minor errors. My comments aimed to enhance the clarity of it.

Line 30 – Please select more carefully keywords. Please avoid the duplication of words already used in the article’s title.

ANSWER: The keywords are corrected

Line 62 – please correct “hem agglutinin – neuraminidase” into “hemagglutinin – neuraminidase”

ANSWER: "hemagglutinin-neuraminidase" is corrected

Line 60 - does writing in bold Peronospora sp. and Phytoftora sp. have any special meaning?

ANSWER: Peronospora sp. and Phytoftora sp. are corrected into nonbold

Line 64 – why in vitro is written bold ?

ANSWER: in vitro is written in nonbold

Line 65 – present experiment instead of “presented”

ANSWER: present experiment is instead "presented"

Line 105 – Why the acronym PBS is expanded for the second time ? (the first one was in line 78). By the way it was used for the first time in line 62 and this should the place of expansion which should looks like “10% phosphate buffer saline (PBS)”

ANSWER: It is written: “10% phosphate buffer saline (PBS)”

Line 86 – Holocene minerals are sometimes abbreviated to HM and sometimes not. Please unify it.

ANSWER: Holocene minerals are unifyed to HM

Line 108 – please correct “HSA (Human Serum Albumin)” into “human serum albumin (HSA)” .

ANSWER: “HSA (Human Serum Albumin)” is corrected into “human serum albumin (HSA)” .

Line 125 – please correct “mine – rals” into “minerals”

ANSWER: "mine-rals" is corrected into "minerals"

Line 134 - please provide country of origin of Human IFN-α

ANSWER: Country or origin of Human IFN-α is USA

Line 150 – there is no need to abbreviate nitrite. It is a simple word. Moreover, NO is commonly used for nitric oxide. Therefore it could be misleading.

ANSWER: Nitric oxide is changed into NO

Line 350 – “Paramyxoviruses” should be written in italic.

ANSWER: "Paramyxoviruses" is in italic.

Line 317 – I see no reason to use references to figures in Discussion. These should be used in Results only.

ANSWER: According to our view the references in Discussion give the seriosity to the text.

Line 318 – why “The Possible Mechanisms” is written in capital letters ?

ANSWER: “The Possible Mechanisms” is changed into "The possible mechanisms"

Line 391, 403, 442, 443  – please correct “Fe++” into “Fe+2

ANSWER: "Fe++" is changed to"Fe+2"

Line 413 – please correct “ROS (Reactive Oxygen Species)” into “Reactive oxygen species (ROS)”

ANSWER: “ROS (Reactive Oxygen Species)”is changed into “Reactive oxygen species (ROS)”

Line 408 - Candida galbrata should be written in italic

ANSWER: Candida galbrata is written in italic.

Line 436 – Why ROS is expanded for the second time ?

ANSWER: ROS is changed without explanation.

All the requested corrections are presented in the corrected article.

Round 2

Reviewer 3 Report

The revised manuscript "the enhancing effects of 10% PBS washout of holocene mine - rals on HuIFN - alpha N3 inducing capacity of NDV ZG1999HDS or sendai virus (Cantrell strain)" includes the requested changes of the reviewer 3. While the experimental design is in general fine, i do miss the correct statistical analysis of the data, e.g. in figure 2 the authors used students t-test instead of Two-way ANOVA followed by a post-test. Please correct this. Students t test is only suggested for figures that include 1 difference in the conditions. In figure 2 the authors used not only different virus strains but also various concentrations of HA/mL which subsequently requires another statistical analysis. In figure 3 the data should be analyzed by One-way ANOVA followed by a post-test.

Please also expand the figure legend and increase the readability.

Please check also the text formation in line 367 and 368.

Please correct in line 398 Nf-B to NFkB and carefully check the manuscript for typing mistakes.

The authors also discussed extensively the effect of ROS on NFkB activation, however, they do not show the effect of the used FeCl2, FeCl3, and KCl on the phosphorylation of NFkB levels in cells.I would strongly encourage the authors to expand this to improve their manuscript.

Author Response

Comments and Suggestions for Authors:

The revised manuscript "the enhancing effects of 10% PBS washout of holocene mine - rals on HuIFN - alpha N3 inducing capacity of NDV ZG1999HDS or sendai virus (Cantrell strain)" includes the requested changes of the reviewer 3. While the experimental design is in general fine, i do miss the correct statistical analysis of the data, e.g. in figure 2 the authors used students t-test instead of One-way ANOVA followed by a post-test. Please correct this. Students t test is only suggested for figures that include 1 difference in the conditions. In figure 2 the authors used not only different virus strains but also various concentrations of HA/mL which subsequently requires another statistical analysis.

ANSWER

In the FIGURE 2 we included the One-way ANOVA followed by a post test. The one-way ANOVA followed by a post test was performed separately for NDV ZG1999HDS and Sendai virus (Cantell strain).

The results are the following:

  1. NDV ZG1999HDS

Summary of Data

Treatments

1

2

3

4

5

Total

N

5

5

5

5

20

∑X

2145

1235

1425

2398

7203

Mean

429

247

285

479.6

360.15

∑X2

923947

309731

406763

1150660

2791101

Std.Dev.

30.5859

34.2272

12.6293

12.0333

101.8101

Result Details

Source

SS

df

MS

Between-treatments

187295.35

3

62431.7833

F = 103.56535

Within-treatments

9645.2

16

602.825

Total

196940.55

19

The f-ratio value is 103.56535. The p-value is < .00001. The result is significant at p < .05.

Post Hoc Tukey HSD (beta)

The Tukey's HSD (honestly significant difference) procedure facilitates pair wise comparisons within your ANOVA data. The F statistic (above) tells you whether there is an overall difference between your sample means. Tukey's HSD test allows you to determine between which of the various pairs of means - if any of them - there is a signficant difference.

A couple of things to note. First, a blue value for Q (below) indicates a significant result. Second, it's worth bearing in mind that there is some disagreement about whether Tukey's HSD is appropriate if the F-ratio score has not reached significance.

Pairwise Comparisons

HSD.05 = 44.4270
HSD.01 = 57.0081

Q.05 = 4.0461    Q.01 = 5.1919

T1:T2

M1 = 429.00
M2 = 247.00

182.00

Q = 16.58 (p = .00000)

T1:T3

M1 = 429.00
M3 = 285.00

144.00

Q = 13.11 (p = .00000)

T1:T4

M1 = 429.00
M4 = 479.60

50.60

Q = 4.61 (p = .02296)

T2:T3

M2 = 247.00
M3 = 285.00

38.00

Q = 3.46 (p = .10781)

T2:T4

M2 = 247.00
M4 = 479.60

232.60

Q = 21.18 (p = .00000)

T3:T4

M3 = 285.00
M4 = 479.60

194.60

Q = 17.72 (p = .0000

  1. Sendai virus (Cantell strain)

Summary of Data

Treatments

1

2

3

4

5

Total

N

5

5

5

5

20

∑X

1253

1214

1628

2831

6926

Mean

250.6

242.8

325.6

566.2

346.3

∑X2

318599

296632

530698

1606915

2752844

Std.Dev.

33.9013

21.6379

12.4619

31.6338

136.5689

Result Details

Source

SS

df

MS

Between-treatments

343276.2

3

114425.4

F = 165.02672

Within-treatments

11094

16

693.375

Total

354370.2

19

The f-ratio value is 165.02672. The p-value is < .00001. The result is significant at p < .05.

Post Hoc Tukey HSD (beta)

The Tukey's HSD (honestly significant difference) procedure facilitates pair wise comparisons within your ANOVA data. The F statistic (above) tells you whether there is an overall difference between your sample means. Tukey's HSD test allows you to determine between which of the various pairs of means - if any of them - there is a significant difference.

A couple of things to note. First, a blue value for Q (below) indicates a significant result. Second, it's worth bearing in mind that there is some disagreement about whether Tukey's HSD is appropriate if the F-ratio score has not reached significance.

Pairwise Comparisons

HSD.05 = 47.6470
HSD.01 = 61.1400

Q.05 = 4.0461    Q.01 = 5.1919

T1:T2

M1 = 250.60
M2 = 242.80

7.80

Q = 0.66 (p = .96494)

T1:T3

M1 = 250.60
M3 = 325.60

75.00

Q = 6.37 (p = .00184)

T1:T4

M1 = 250.60
M4 = 566.20

315.60

Q = 26.80 (p = .00000)

T2:T3

M2 = 242.80
M3 = 325.60

82.80

Q = 7.03 (p = .00072)

T2:T4

M2 = 242.80
M4 = 566.20

323.40

Q = 27.46 (p = .00000)

T3:T4

M3 = 325.60
M4 = 566.20

240.60

Q = 20.43 (p = .00000)

In figure 3 the data should be analyzed by One-way ANOVA followed by a post-test.

ANSWER

The data in the FIGURE 3 and in the FIGURE 4 are so complex, which we did not, performed by the one way ANOVA followed by a post test.

Please also expand the figure legend and increase the readability.

ANSWER

We have expanded the FIGURE 3 and FIGURE 4 legend and so increased the readability.

Please check also the text formation in line 367 and 368.

ANSWER

We have checked the text formation in the line 367 and 368 as follows: The tumour selectivity of lentogenic NDVs is considered to be due to a defective IFN’s response in tumour cells. Moreover, the strain NDV ZG1999HDS is a relatively strong inducer of human type I IFNs, more precisely of the HuIFN-αN3, in the PBMCs from human Buffy coats.

Please correct in line 398 Nf-B to NFkB and carefully check the manuscript for typing mistakes.

ANSWER

In the line 398 the Nf-B is corrected to the NFkB. The manuscript is caferully checked for the typing mistakes.

The authors also discussed extensively the effect of ROS on NFkB activation, however, they do not show the effect of the used FeCl2, FeCl3, and KCl on the phosphorylation of NFkB levels in cells.I would strongly encourage the authors to expand this to improve their manuscript.

ANSWER

Iron exacerbates various types of liver injury in which nuclear factor NF-κB-driven genes are implicated. This study tested a hypothesis that iron directly elicits the signaling required for activation of NF-κB and stimulation of tumor necrosis factor (TNF)-α gene expression in Kupffer cells. Addition of Fe2+(FeCl2) but not Fe3+(FeCl3) (∼5–50 μM) to cultured rat Kupffer cells increased TNF-α release and TNF-α promoter activity in a NF-κB-dependent manner. Fe2+ (FeCl2) caused a disappearance of the cytosolic inhibitor κBα, a concomitant increase in nuclear p65 protein, and increased DNA binding of p50/p50 and p65/p50 without affecting activator protein-1 binding. Addition of Fe2+ (FeCl2) to the cells resulted in an increase in electron paramagnetic resonance-detectable ·OH peaking at 15 min, preceding activation of NF-κB but coinciding with activation of inhibitor κB kinase (IKK) but not c-Jun NH2-terminal kinase. In conclusion, Fe2+ (FeCl2)serves as a direct agonist to activate IKK, NF-κB, and TNF-α promoter activity and to induce the release of TNF-α protein by cultured Kupffer cells in a redox status-dependent manner. We propose that this finding offers a molecular basis for iron-mediated accentuation of TNF-α-dependent liver injury.

This manuscript is a resubmission of an earlier submission. The following is a list of the peer review reports and author responses from that submission.

Round 1

Reviewer 1 Report

The study has described interesting findings that the strains of NDV ZG1999HDS and SV (Cantell strain) can induce production of HuIFN-αN3 from PMBC, which can be enhanced with 10% HM-PBS washout of Holocene minerals. The manuscript needs slightly revise before being accepted for publication.

1 Based on the data from Figure2, the amount of the viruses does not have a linear correlation with their induction capacity of HuIFN-αN3. Why do the authors select the concentration of 100 HA/ml not other to further study together with 10% HM-PBS ?

2 Regarding the statistics analysis, it should be better to add the number of replication for each assay as format n=3 or n=4 in appropriate position of the text, figure and table. It should also be better to mark the difference whether it is significant or not in 95% level between different group.

3 On Holocene minerals mentioned at line 78, give a brief description where the study got it.

4 The following typos should be corrected.

line 42, gens--genes

line 126 »Platinum ELISA«

Table 4 and others

3.818.21 ± 41.9-----3,818.21 ± 41.9

4.790.34 ± 33.5---4,790.34 ± 33.5

2.695.10 ± 22.4---2,695.10 ± 22.4

3.447.29±47.3---3,447.29±47.3

Reviewer 2 Report

The manuscript have many grammar and spelling mistakes and needs extensive English editing

Author Response

The manuscript has many grammar and spelling mistakes and needs extensive English editing

Response: The manuscript was checked and corrected for grammar and spelling mistakes. (See Corrected Life - 1445830)